# Development of the Ward Nurses' Perspective-taking of the Staff Receiving Discharged Patients Scale: An observational study of ward nurses

Shingo Tanaka[1]*, Masatoshi Saiki[2], Yukie Takemura[3]

1 Department of Fundamentals Nursing, Nursing Course, School of Medicine, Yokohama City University, Kanagawa, Japan, 2 Department of Advanced Clinical Nursing, Frontier Clinical Nursing, Graduate School of Nursing, Chiba University, Chiba, Japan, 3 Nursing Department, The University of Tokyo Hospital, Tokyo, Japan

* tanaka.4n5.res@gmail.com

**Data Availability Statement:** All relevant data are within the paper and its Supporting Information files.

## Abstract

### Introduction

Discharge planning involves coordinating between different care settings. Failed coordination can lead to delayed care at the facilities receiving discharged patients. Nurses who implement discharge planning must consider the circumstances of the staff receiving the discharged patients.

### Objective

This study aims to develop a nurses' perspective-taking scale for measuring the cognitive process of imagining a staff's situation when receiving discharged patients.

### Methods

An online survey was conducted from September to November 2021 with nurses in Japanese acute care hospitals, using a 20-item questionnaire based on interviews and a literature review. Item reduction was conducted based on inter-item correlations, item-total correlations, and exploratory factor analysis. Confirmatory factor analysis was performed. For the exploratory and confirmatory factor analysis, the samples were randomly split in half and tested. Cronbach's alpha, intraclass correlation coefficients, and correlation coefficients were calculated using both self-developed and pre-existing measures.

### Results

The questionnaire was distributed to 1,289 individuals, and 416 valid responses were obtained. Item reduction and exploratory factor analysis resulted in a 10-item scale with two subscales, "imagine-other" (six items), which is imagining the other person's situation, and "imagine-self" (four items), which is imagining oneself in the other person's situation. The goodness-of-fit indices were as follows: comparative fit index, 0.95; root mean square error

**Funding:** •ST. •There is no grant number. •The Global Creative Leaders (GCL) program of The University of Tokyo by the Ministry of Education, Culture, Sports, Science and Technology (MEXT). •https://www.gcl.i.u-tokyo.ac.jp/faq-2/subsidies/ •The sponsors or funders did not play any role in the study.

**Competing interests:** The authors declare that there is no conflict of interest.

of approximation, 0.08; and standardized root mean squared residual, 0.06. Correlation coefficients with existing scales were 0.38, 0.57, and -0.33. Cronbach's alpha was 0.89, and the intraclass correlation coefficient was 0.54.

## Conclusion

The newly developed scale proved to be reliable, valid, and suitable for use. This scale can measure the degree of perspective-taking by nurses, which can improve collaboration between facilities and the effectiveness of discharge planning.

## Introduction

Discharge planning involves sequential processes, including patient education during the hospital stay and post-discharge care coordination. Ward nurses play a crucial role in planning, coordinating multidisciplinary teams, and educating patients [1, 2]. High-quality discharge planning is needed to ensure continuity of care even after leaving the facility.

Fragmentation of care across acute, post-acute, or home care settings leads to poor discharge planning coordination and delays in care delivery [3, 4]. Nurses' perceptions of discharge planning vary across different healthcare facilities, leading to dysfunctional inter-facility cooperation systems [5, 6]. This may be due to the wide range of resources, the focus on patients in each care setting, and a lack of consideration for these differences during discharge planning [5, 7, 8]. Improving discharge planning requires enhancing both systems and nurses' recognition of resources and perspectives of care, depending on the facilities. Proficient discharge planners think beyond their own facilities to ensure continuity of care [9]. To improve the quality of discharge planning, assessing how well nurses in pre-discharge facilities can anticipate situations at post-discharge facilities is essential.

Perspective-taking is the cognitive ability to imagine others' viewpoints, developed during social interactions [10]. It is a subset of empathy and is crucial for understanding others [11]. It involves imagining the world from another's perspective—comprehending their viewpoint, thoughts, motivations, intentions, and emotions [12]. Studies in developmental and social psychology state that it includes de-centering and does not require accurate understanding of the other's perspective [12]. Perspective-taking has two sub-concepts: imagine-other for objective understanding and imagine-self for subjective understanding by placing oneself in another's situation [13]. The use of these two sub-concepts is realistic as it considers objects both subjectively and objectively.

In healthcare, research on perspective-taking focuses on healthcare professionals' empathy for patients [14]. Psychology and management studies link perspective-taking with cooperative and helping behaviors [12, 15, 16]. In nursing, understanding the perspective of staff receiving discharged patients can enhance collaboration. The most widely used perspective-taking scale is the Davis's Interpersonal Reactivity Index [11]. However, no current measures assess perspective-taking for staff receiving discharged patients [12, 17–19]. Thus, there is a need to develop a new instrument to measure this aspect at the discharge planning stage and to test its reliability and validity.

Prior research on discharge planning has focused on nurses' recognition and knowledge of patients and their families after discharge [20, 21]. As discharge planning requires collaboration with the staff at the post-discharge facility, examining nurses' perceptions of them is essential. Scale development allows us to measure the extent to which nurses consider the

staff's situation at receiving facilities. The developed scale can help examine the relationship between nurses' perspective-taking and discharge planning outcomes, contributing to enhancing nurses' discharge planning competencies. Therefore, this study develops a scale to measure ward nurses' perspective-taking of the staff receiving discharged patients (Ward Nurses' Perspective-taking of the Staff Receiving Discharged Patients Scale) and tests the scale's reliability and validity.

## Methods

This study followed Boateng et al.'s [22] three steps of scale development: item development (literature review and interview), expert evaluation, and scale validation.

### Definition

In this study, the staff receiving discharged patients comprise nurses, home care nurses, and public health nurses in the jurisdictional district who receive patients from the hospital after discharge. Referring to previous studies (e.g., Ku et al. [12]), ward nurses' perspective-taking of the staff receiving discharged patients was defined as the active cognitive process of imagining the perspectives, positions, and situations of the staff receiving discharged patients when ward nurses implement discharge planning (including transfer to another hospital and/or home-care) for patients requiring continued medical care.

### Three steps of item development

**Item development.** The item was created in three steps: 1) establishing sub-concepts, 2) creating items through a literature review and interviews, 3) synthesizing the created items. First, drawing from Wolgast et al. [13], we assumed two sub-concepts of perspective-taking: imagine-other—imagining the other person's situation—and imagine-self—imagining oneself in the other person's situation. Second, through a literature review and interviews, we collected data on what nurses consider about the discharge facility during discharge planning and created a draft of items. For the literature review, the following terms were used to search the CINAHL and CiNii databases for relevant articles: "transitional care," "transition of care," "discharge planning," and "nurse." Based on the results of the literature review [3, 4, 21, 23–27], items such as "We try to imagine how staff receiving discharged patients value the care we give them" (28 items in total) were developed. We then conducted individual semi-structured interviews with nine participants, including ward nurses, discharge-coordinating nurses, and ward nursing managers with at least five years of experience working in hospital wards and discharge planning (mean clinical experience year 16.66±10.75). Purposive sampling was used for recruitment. We asked them questions such as, "To achieve smooth discharge planning, what, in particular, do you consider about the situation of the staff receiving discharged patients?" The extent to which the participants imagine the perspective of the staff receiving discharge patients was ascertained through these methods. Based on the interview results, items such as "Consider the amount of care that can be handled by the staff receiving discharged patients" (17 items in total) were created. Third, items created using literature review and interview data were summarized among those with similar content. "Overall, 28 items (12 imagine-other and 16 imagine-self items) were included based on the abovementioned steps.

**Evaluation by experts.** In addition to the nine participants interviewed in the item development phase, cognitive interviews [28] were conducted with five ward nurses with discharge planning experience to evaluate surface validity. Additionally, the item content validity index (I-CVI) was calculated to test content validity through nursing researchers' and nurses' evaluations [29, 30]. The cognitive interview and I-CVI were conducted twice alternately. Each item

was modified accordingly (see Supporting information). The inclusion criteria were participants with clinical experiences of at least 5 years as a nurse and in discharge planning. The exclusion criteria included only having clinical experience in the maternity ward.

In the cognitive interviews, based on their clinical experience, participants were asked about questions that were difficult to understand and ones that should be included.

The I-CVI was evaluated by four nursing researchers and four nurses who were deemed by the researcher to understand perspective-taking in the interviews in the item development phase. Eight participants were evaluated in the first round and seven in the second round (one nursing researcher dropped out). Items with an I-CVI value less than 0.75 (eight participants) in the first round and less than 0.83 (six participants) in the second round were considered for modification or deletion. The criteria for each round were calculated based on the recommendation of a previous study [29].

In the first round, subject and predicate expressions were standardized in response to the cognitive interview results. The I-CVI was calculated for the revised draft, and items that did not meet the criteria were revised and synthesized based on the comments obtained from the cognitive interview and the I-CVI [29]. Through this process, the items were rephrased, integrated with other items, or deleted, resulting in 22 items—down from the original 28. Items with I-CVI values that did not meet the criteria in the second round were similarly reviewed by the researchers and deleted in consideration of feedback from the cognitive interviews. This resulted in 20 items. The researchers reviewed the final draft to confirm the semantic content, and minor phrasing revisions were made to facilitate easy responses from participants. Overall, 20 items were refined through this process (9 imagine-other and 11 imagine-self items).

**Scale validation.** An online survey was conducted from October to November 2021. The participants were given individual random IDs and surveyed again two weeks later, if they cooperated, for temporal stability verification.

**Data collection of scale validation.** *Sample size*. We performed sample size calculations to validate construct validity at time 1 and temporal stability at time 2. Regarding time 1, we observed that 200 participants, 10 times the number of items in the draft scale, were required based on the recommendation of Boateng et al. [22]. Regarding time 2, we found 19 participants were needed to calculate the intraclass correlation coefficients (ICC) between the first and the second surveys with 30 observed variables per participant (the null hypothesis was set at 0.5, the alternative hypothesis at 0.7, the power at 0.8, and the probability of significance at 0.05) [31]. The response rate was assumed to be approximately 20% based on previous discharge planning studies [32, 33]. Therefore, it was determined that a distribution of approximately 1,000 persons was necessary for the analysis.

*Surveyed facilities*. Overall, 45 facilities that agreed to participate were selected through random sampling of acute care hospitals without medical care beds in the Kanto Koshinetsu, Tohoku, and Tokai Hokuriku Health and Welfare Bureaus areas (selected based on facility reports issued by the Kanto-Shinetsu Regional Bureau of Health and Welfare [34], Tohoku Regional Bureau of Health and Welfare [35], and Tokai-Hokuriku Regional Bureau of Health and Welfare [36]) in Japan). Random sampling involved randomly extracting 200 acute care hospitals from databases provided by each local health authority [34–36] and requesting their cooperation for the survey. Of these, 45 facilities that agreed to participate were selected for the survey.

*Participants*. The inclusion criteria were ward nurses who had engaged in discharge planning. The exclusion criteria were nurses with less than a year of nursing experience, did not work in wards, worked in obstetric wards and unit care such as intensive care units and emergency wards, and worked as head ward nursing managers.

**Ethical considerations.** This study was conducted with the approval of the Research Ethics Committee of The University of Tokyo, Clinical Research Review Board (approval no. 2020409NI, 2021137NI). Participants were informed that their cooperation in the survey was voluntary and that their responses would not be used to evaluate their work. Furthermore, it was clarified that their participation and the content of their responses would not be shared with any third party. The survey was administered only to individuals who willingly agreed to participate and provided written informed consent.

## Data analysis

**Item reduction.** Items that were similar or did not adequately measure perspective-taking were removed based on the following criteria: 1) inter-item correlations greater than 0.60 [22]; 2) item total (IT) correlations less than 0.30 [22]; 3) factor loadings in the exploratory factor analysis (EFA) of 0.30 or greater for multiple factors [37]; 4) commonality less than 0.4 in the EFA [38].

**Determination of factor structure.** To test for cross-validity, the collected sample was randomly divided into two groups. EFA was conducted for one group and confirmatory factor analysis (CFA) for the other. This scale was designed with a multidimensional structure in mind. Therefore, EFA was employed to identify and name the factors from the item pool. Subsequently, CFA was conducted to verify the construct validity by examining whether the model fit statistically across different samples. In the EFA, the number of factors was determined by scree plotting; subsequently, Promax rotation and unweighted least squares were selected. Since a Shapiro–Wilk test of the items did not confirm normality, the unweighted least squares method was selected. The comparative fit index (CFI), root mean square error of approximation (RMSEA), and standardized root mean squared residual (SRMR) were used as the evaluation indices for the CFA.

**Scale evaluation.** Internal consistency was confirmed by Cronbach's alpha. The criterion for the index was set at 0.7 [39].

Construct validity was verified by calculating correlations between the Ward Nurses' Perspective-taking of the Staff Receiving Discharged Patients Scale and similar concepts [40]. Details of the scales used for validation and the predicted results were as follows: 1) The Japanese version of the Interpersonal Reactivity Index, perspective-taking subscale [11, 41], measures perspective-taking in daily life and therefore although similar, has a slightly different concept from the Ward Nurses' Perspective-taking of the Staff Receiving Discharged Patients Scale. The correlation was predicted to be weak to moderate. 2) Since there was no golden standard scale to measure comprehensive perspective-taking in discharge planning, we predicted that the Ward Nurses' Perspective-taking of the Staff Receiving Discharged Patients Scale would have a moderate or high correlation with the self-created single item (named "comprehensive perspective-taking") "I think about discharge support from the perspective of the receiving staff." The participants were asked to respond to this statement on a 10-point scale ranging from 1 = *not at all true* to 10 = *very true*. 3) The sub-concept of the Self-centeredness Scale, also known as "inadequate empathy for others," was developed by Hirosawa et al. [42] based on Piaget's [43] concept of self-centeredness. Self-centeredness and de-centering are paired concepts, and perspective-taking is a de-centering concept [11]. Therefore, a negative correlation between perspective-taking and "inadequate empathy for others" is expected.

The ICC (2, 1) [44] was calculated to verify temporal stability. The online survey did not have any missing data because participants were unable to submit the survey unless they answered all the questions. For statistical analysis, IBM SPSS Statistics (Ver. 28.0.0 [190]) and IBM SPSS Amos (Ver. 25.0.0) were used. The significance level was set at 5%.

# Results

## Participants

For the first survey (Time 1), the questionnaire was distributed to 1,289 participants, and 416 valid responses were obtained (Fig 1). For the second survey (Time 2), questionnaires were distributed to 172 participants who participated in the first survey and offered to cooperate in the second—61 valid responses were obtained.

In the sample surveyed at Time 1 ($n$ = 416), 93.8% were women, the mean age was 38.90 years, and the mean clinical experience was 15.62 years. Of the total, 67.1% of the participants held a staff position, and 44.2% had participated in discharge planning at least twice a month in the previous year (Table 1).

## Scale validity

In the item reduction phase, no item had an IT correlation of 0.3 or less. Of items with an IT correlation of 0.6 or higher and with overlapping meanings, seven with abstract questions were eliminated. The EFA conducted at this stage confirmed a three-factor structure, unlike the assumed factor structure. Three items had factor loadings of 0.3 or higher on multiple sub-factors and were deleted because the questions were documents asking about multiple events. No item had a commonality of less than 0.4. Ten items were removed through the item reduction process. The EFA conducted at this stage confirmed a two-factor structure consisting of items that asked about the assumed sub-concepts (Table 2). The first factor was named "imagine-other" (six items), and the second factor was named "imagine-self" (four items).

The CFA of the factor structure validated by the EFA yielded a CFI of 0.95, an RMSEA of 0.08, and an SRMR of 0.06 (Fig 2).

## Scale reliability

Cronbach's alpha for the Ward Nurses' Perspective-taking of the Staff Receiving Discharged Patients Scale, calculated as the sum of the means of the subscales, was 0.89. The correlation coefficients between the Ward Nurses' Perspective-taking of the Staff Receiving Discharged Patients Scale and the Japanese version of the Interpersonal Reactivity Index subscales of "perspective-taking" (excluding reversal items), "inclusive perspective-taking," and "inadequate empathy for others" were 0.38, 0.57, and -0.33, respectively (Table 3).

The ICCs (2, 1) for the Ward Nurses' Perspective-taking of the Staff Receiving Discharged Patients Scale and the imagine-other and imagine-self subscales were 0.50, 0.62, and 0.24, respectively. Corresponding $t$-tests exhibited significantly higher mean scores for the Ward Nurses' Perspective-taking of the Staff Receiving Discharged Patients Scale and the imagine-self subscale in Time 2 than in Time 1 (Table 4).

# Discussion

The reliability and validity of the scale developed in this study to measure ward nurses' perspective-taking of the staff receiving discharged patients were verified. The EFA and CFA confirmed that the scale consists of two subfactors. Correlation analysis with other scales confirmed its construct validity, whereas Cronbach's alpha and ICC confirmed its reliability. However, the "imagine-self" subfactor was observed to have low temporal stability.

## Participant attributes

The participants' average years of clinical experience were similar to those of previous studies on nurses' discharge planning practices [45]. The most common frequency of involvement in

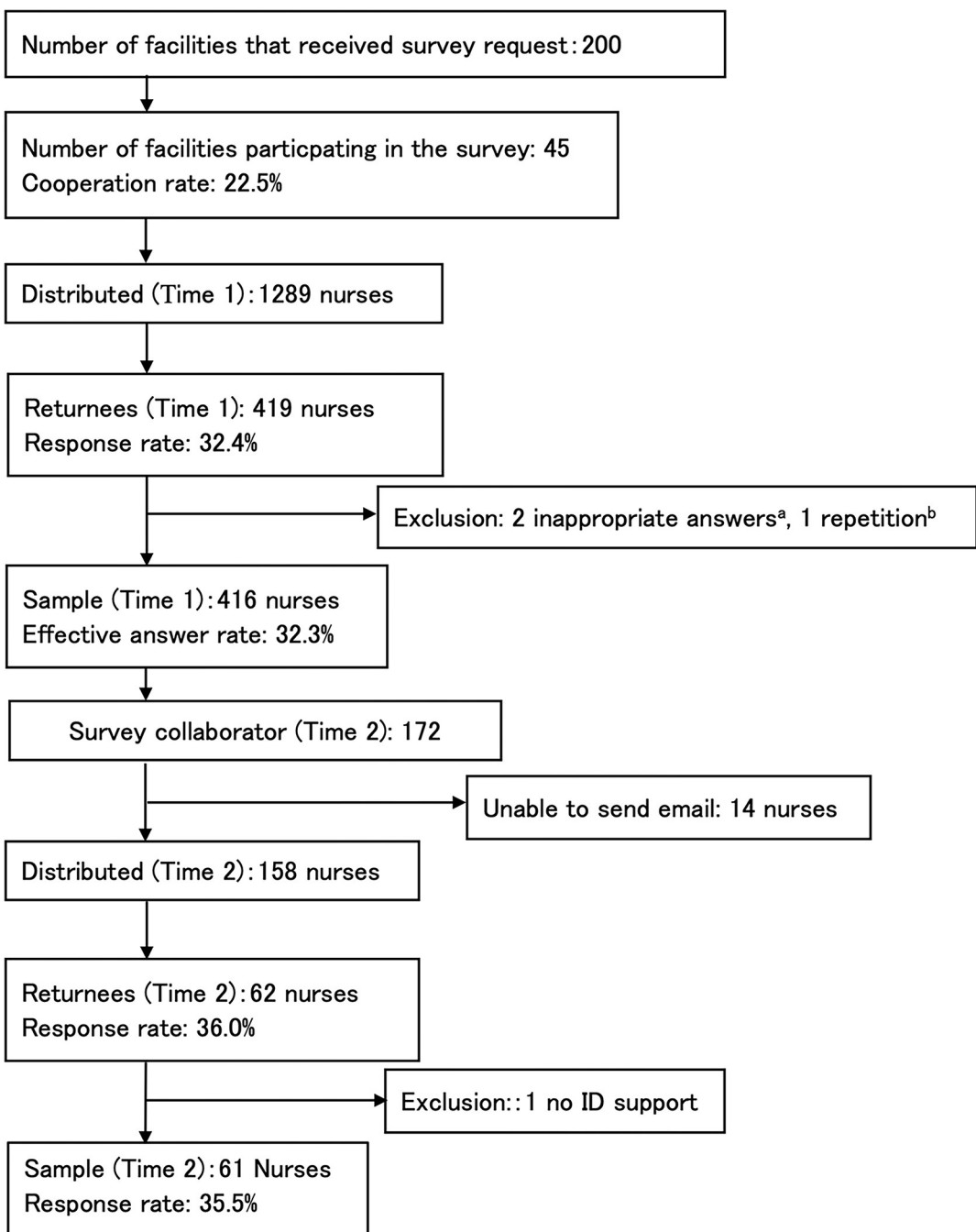

**Fig 1. Flowchart of the process of sample extraction.** *Note*. [a] In the middle of the Ward Nurses' Perspective-taking of the Staff Receiving Discharged Patients Scale, an opposite item, "I do not consider the receiving staff's situation when providing discharge support" was included to determine inappropriate answers. Items that met the following two conditions were excluded: 1) a response score of 1 or 5 on the opposite item; 2) 51% or more of the response items on the Ward Nurses' Perspective-taking of the Staff Receiving Discharged Patients Scale were the same as the opposite item. There were two applicable responses. [b] Since there were two responses with the same ID and the basic attributes matched, we considered them as duplicates and adopted the latest response. [c] Those who agreed to cooperate in the second survey (Time 2) among the respondents of the first survey (Time 1).

**Table 1. Basic attributes of the participants (*n* = 416).**

| | | Mean | (Standard deviation) |
|---|---|---|---|
| **Age** | | **38.90** | **(9.49)** |
| **Years of clinical experience** | | **15.62** | **(9.25)** |
| | | **Frequency** | **(%)** |
| Gender | | | |
| | Female | 390 | (93.75) |
| | Male | 26 | (6.25) |
| Marital status | | | |
| | Married | 224 | (53.85) |
| | Unmarried | 192 | (46.15) |
| Certified nurse/certified nurse specialist | | | |
| | Yes | 16 | (3.85) |
| | No | 400 | (96.15) |
| Current position | | | |
| | Staff | 279 | (67.07) |
| | Manager (Deputy chief, Chief, Deputy head ward nursing manager)[†] | 137 | (32.93) |
| Main department of the ward to which the participant currently belongs | | | |
| | Surgery | 96 | (23.07) |
| | Internal medicine | 161 | (38.70) |
| | Mixed surgery and internal medicine | 151 | (36.30) |
| | Psychiatry and pediatrics[‡] | 8 | (1.92) |
| Number of times in the past year that the participant has been involved in assisting in the discharge of patients who continue to receive care at another facility after discharge | | | |
| | Never | 20 | (4.81) |
| | Approximately once every 6 months | 21 | (5.05) |
| | About once every 2 to 3 months | 90 | (21.63) |
| | About once a month | 101 | (24.28) |
| | About twice a month | 184 | (44.23) |

Note.

[†] 16 deputy directors, 83 directors, 38 deputy chief nursing officers.

[‡] 3 psychiatry, 5 pediatrics.

discharge support in the past year was "at least twice a month" (44.2%). Prior studies have reported that nurses engage in patient education [46] and the coordination of discharge planning [2, 47]. Furthermore, this study's results suggest that most nurses engage in discharge support.

## Scale validity

The EFA extracted two factors, as expected. Factor 1 included a convergence of items that inferred the situation of the staff receiving discharged patients and corresponded to the predicted "imagine-other" factor. Factor 2 included items that asked participants to imagine themselves in the position of the staff receiving discharged patients. These questions were converged in Factor 2, which corresponded to the predicted "imagine-self." The CFA revealed that

**Table 2. Results of exploratory factor analysis (*n* = 208).**

| Ward Nurses' Perspective-taking of the Staff Receiving Patients in Discharge Planning Scale (α = 0.90) | Factor 1 | Factor 2 |
|---|---|---|
| **Factor 1: Imagine-other (α = 0.89)** | | |
| S5 I think about the daily-living care that the staff receiving discharged patients might be able to handle. | 0.95 | -0.12 |
| S4 I think about the manpower of the facility receiving the discharged patients. | 0.79 | -0.07 |
| S7 I think about what care items are available at the site of the staff receiving discharged patients. | 0.73 | 0.04 |
| S6 I think about possible procedures and other medical techniques that can be performed by the staff receiving discharged patients. | 0.73 | 0.09 |
| S8 I think about what information the staff receiving discharged patients already has about the patient. | 0.62 | 0.11 |
| S2 I imagine what the staff receiving discharged patients would want from our discharge planning. | 0.60 | 0.13 |
| Factor 2: Imagine-self (α = 0.87) | | |
| S12 I think about what information I would need to manage a patient's condition if I were the staff receiving discharged patients. | -0.14 | 0.99 |
| S13 I think about how I would understand the information provided by the hospital where the receiving-patient was located if I were the staff receiving discharged patients. | 0.05 | 0.83 |
| S11 I think about what patient family information I would need if I were the staff receiving discharged patients. | 0.08 | 0.72 |
| S15 I think about whether I would need to be careful when interacting with the patient if I were the staff receiving discharged patients. | 0.30 | 0.49 |
| Eigenvalue | 5.43 | 1.38 |
| Factor contribution rate (%) | 50.49 | 10.75 |

*Note*. Kaiser-Meyer-Olkin's Measure of Sample Adequacy was 0.899. Bartlett's Sphericity test had a probability of significance of less than 0.01 (approximate Chi-Square = 1196.90, d.f. = 45). Factor extraction was performed by the unweighted least squares method. The rotation method was Promax rotation. Three iterations were used for convergence. α shows Cronbach's α calculated with *n* = 208. This scale is a 5-point Likert scale, ranging from "1: Not at all applicable" to "2: Not very applicable," "3: Can't say," "4: Somewhat applicable," and "5: Very applicable".

CFI ($\geq$ 0.90; [39, 48]), RMSEA (< 0.10; [39, 49]), and SRMR (< 0.80; [50]) met the criteria. Consequently, the Ward Nurses' Perspective-taking of the Staff Receiving Discharged Patients Scale was established as consisting of the subscales imagine-other and imagine-self.

Construct validity was assessed by examining correlations with three measures: 1) The Ward Nurses' Perspective-taking of the Staff Receiving Discharged Patients Scale correlated weakly with the "perspective-taking" subscale of the Japanese version of the Interpersonal Reactivity Index [11, 41], as predicted. 2) "Comprehensive perspective-taking," which asked about the overall perspective-taking of the staff receiving discharged patients in discharge planning, was moderately correlated with the Ward Nurses' Perspective-taking of the Staff Receiving Discharged Patients Scale, as predicted. 3) The measure of self-centeredness, "inadequate empathy for others," was negatively correlated with the Ward Nurses' Perspective-taking of the Staff Receiving Discharged Patients Scale, as predicted.

Based on the cross-validity, EFA, CFA, and construct validity results, the Ward Nurses' Perspective-taking of the Staff Receiving Discharged Patients Scale is considered a valid instrument. It measures nurses' perceptions of the resources available to the staff receiving discharged patients and the information they desire. Previous scales have measured perspective-taking toward friends in everyday life, staff in other departments of a company, and customers [11, 17, 18]. However, the scale developed in this study differs from other existing ones

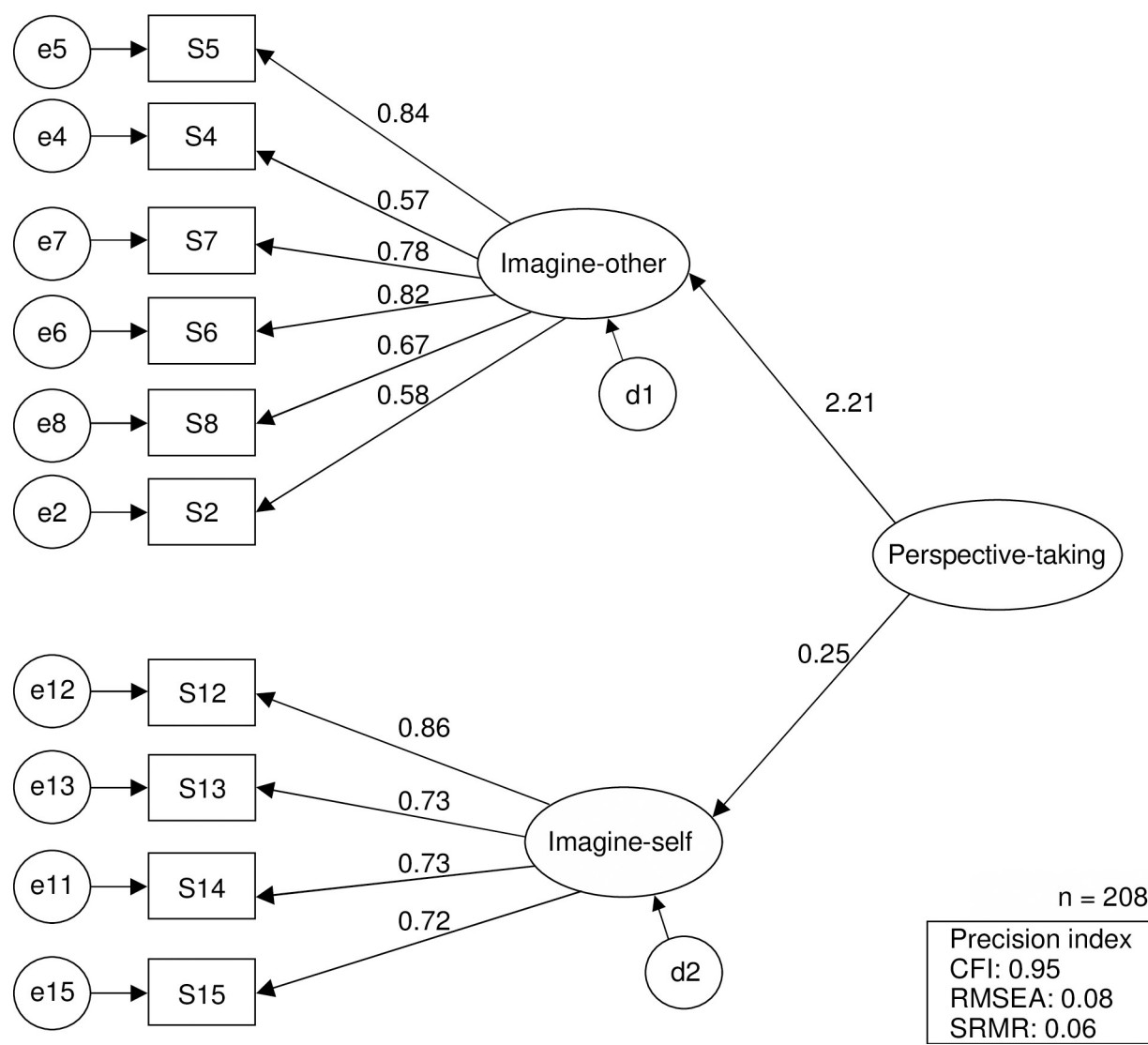

**Fig 2. Model evaluation of the Ward Nurses' Perspective-taking of the Staff Receiving Discharged Patients Scale using secondary factor analysis.** *Note.* For cross-validation, two groups of 208 samples were randomly selected from 416 samples. One group was used for exploratory factor analysis and the other for confirmatory factor analysis. CFI = 0.95; RMSEA = 0.08; SRMR = 0.06.

in that it measures perspective-taking toward staff in other facilities in the medical setting of discharge planning.

## Scale reliability

The Cronbach's alpha of the Ward Nurses' Perspective-taking of the Staff Receiving Discharged Patients Scale met the criterion (> 0.8: good), which indicated internal consistency [39]. The ICCs (2, 1) for the Ward Nurses' Perspective-taking of the Staff Receiving Discharged Patients Scale and the imagine-other subscale were 0.50 and 0.62, respectively, and these values met the criterion of moderate reliability ($0.5 \leq$ ICC $\leq 0.75$; [44]). Conversely, the ICC for the imagine-self subscale was 0.24, which indicated poor reliability. The mean scores for the Ward Nurses' Perspective-taking of the Staff Receiving Discharged Patients Scale and imagine-other were significantly higher in Time 2 than in Time 1. Perspective-taking in daily life has also

**Table 3. Descriptive statistics, Cronbach's alpha, and correlation coefficient for each scale (*n* = 416).**

| | Mean | Standard deviation | (α) | Correlation ② | ③ | ④ | ⑤ | ⑥ |
|---|---|---|---|---|---|---|---|---|
| ①Ward Nurses' Perspective-taking of the Staff Receiving Discharged Patients Scale (10 items)[a] | 7.21 | (1.16) | (0.89) | 0.90** | 0.86** | 0.38** | 0.57** | -0.33** |
| ②Imagine-other (6 items)[b] | 3.34 | (0.72) | (0.88) | | 0.55** | 0.34** | 0.59** | -0.32** |
| ③Imagine-self (4 items)[b] | 3.87 | (0.59) | (0.86) | | | 0.32** | 0.40** | -0.25** |
| ④Perspective-taking (7 items)[c] | 3.65 | (0.60) | (0.78) | | | | 0.35** | -0.42** |
| ⑤Comprehensive perspective-taking (1 item)[d] | 5.99 | (1.68) | | | | | | -0.32** |
| ⑥Inadequate empathy for others (7 items)[e] | 2.51 | (0.72) | (0.84) | | | | | |

*Note*. Correlation coefficients were calculated using Pearson's correlation. ** $p \leq 0.01$.

[a] Scores for each subscale (imagine-other and imagine-self) were summed.

[b] The average score for each item was calculated.

[c] The perspective-taking scale was calculated using the perspective-taking subscale (7 items) of the Japanese version of the Interpersonal Reactivity Index [44], which had a low α = 0.67; therefore, we followed previous research and excluded the reversal items.

[d] A self-developed item, "I think about discharge support from the perspective of the receiving staff," was created and an index was used to rate the item on a 10-point scale, from "1 = not applicable at all" to "10 = applicable very often."

[e] The Self-Centeredness Scale [42], a subscale of the "inadequate empathy for others" scale, was used.

been reported to change over time [51]. Therefore, nurses' perspective-taking of the staff receiving discharged patients may also change over time. In experimental studies, perspective-taking is enhanced by instructing participants to think from the perspective of the object [12]. As such, the responses to the first survey may have increased the scores in the second survey because the participants became unconsciously more aware of perspective-taking. In particular, individuals may become more conscious of their imagined others by teaching or answering a questionnaire. Further studies should be conducted on 1) how nurses' perspective-taking of staff receiving patients changes over time, 2) whether an educational effect was seen in responding to the scale twice, and for how long this effect remains on the nurses.

## Scale availability

The scale developed in this study measures the degree to which ward nurses conducting discharge planning consider the situation of the staff receiving discharged patients. Previously, measures for discharge planning were limited to categories such as psychological readiness [52] and discharge planning behavior [53] as the outcome. Prior research on nurses' perceptions focused on their interest in patients [20, 21]. Additionally, studies on collaboration among medical staff in discharge planning focused on systems and the physical environment

**Table 4. ICC (2, 1) values for the Ward Nurses' Perspective-taking of the Staff Receiving Discharged Patients Scale and paired-samples T-test results (*n* = 416).**

| | ICC (2, 1) Intraclass correlation | 95% CI Lower | Upper | Paired-samples *t* test mean Time 1 | Time 2 | Mean difference[a] | *p* value |
|---|---|---|---|---|---|---|---|
| Ward Nurses' Perspective-taking of the Staff Receiving Discharged Patients Scale | 0.50 | 0.27 | 0.67 | 7.45 | 7.94 | 0.49 | 0.02 |
| Imagine-other | 0.62 | 0.39 | 0.76 | 3.43 | 3.72 | 0.29 | <0.01 |
| Imagine-self | 0.24 | 0.00 | 0.46 | 4.02 | 4.22 | 0.20 | 0.06 |

*Note*. ICC, intraclass correlation coefficients; CI, confidence interval. [a] The mean of Time 2 minus that of Time 1

[54, 55]. Therefore, the manner in which nurses' perceptions of the staff receiving discharged patients would affect their collaborations was unclear. The measures developed by this study focus on nurses' recognition and contribute to clarifying the mechanism whereby the kind of event stimulates the perspective-taking and affects discharge planning [20, 21, 54, 55]. By evaluating perspective-taking, identifying intervention points to improve nurses' perspective-taking during discharge planning will be possible.

## Limitations

The current study was conducted with nurses in acute care hospitals in Japan, which affects the generalizability of its findings. The viability of the scale for nurses other than ward nurses, such as those involved in advanced discharge planning [56], needs to be examined further. The current study met the sample size requirements for time 1 and time 2 [22, 31]. However, the response rate was low, similar to the response rate in another Japanese study of discharge planning [32, 33]. The low response rate may be due to the many questions and participants' interest. In particular, a selection bias may have occurred if only nurses who expressed a strong interest in discharge planning were assumed to have responded to the study questions.

In the present study, the majority of the participants were female. It has been reported that gender does not affect perspective-taking [18]. Therefore, a bias due to gender is unlikely. However, perspective-taking is affected by the power relationship with the target [57, 58] as well as differences between competitive and cooperative environments [16]. Therefore, future research should not only focus on the attributes of the participant's population, but also on the relationship between the perspective-taking target and participant population.

In addition, the imagine-self subscale demonstrated low temporal stability; therefore, there is a limitation to using this subscale by itself. Future longitudinal studies are needed to understand the nature of this subscale and explore avenues to recognize temporal trends in its scores.

Perspective-taking does not question the accuracy of understanding the object's perspective [12]. Consequently, the ideas derived through perspective-taking do not necessarily correspond to what the target individual thinks or the actual situation in which that individual is placed. Future studies should evaluate the relationship between perspective-taking and inter-facility collaboration in detail. To do so, researchers should ask staff receiving discharged patients to evaluate the discharge planning in which the perspective-taking was demonstrated, using the Ward Nurses' Perspective-taking of the Staff Receiving Discharged Patients Scale.

## Conclusions

We developed the Ward Nurses' Perspective-taking of the Staff Receiving Discharged Patients Scale in order to evaluate ward nurses' perspective-taking of the staff receiving discharged patients. The scale comprised 10 items, two factors, and two subscales: imagine-other (six items) and imagine-self (four items). Furthermore, the reliability and validity of this scale and its subscales were verified in this study. The scale measures ward nurses' perspective-taking of the staff receiving discharged patients and allows us to elucidate the relationship between perspective-taking and inter-facility collaboration in discharge planning. This scale can be utilized to measure the extent to which ward nurses engage in perspective-taking to better understand the staff at the facility receiving discharged patients. It can help verify the relationship between perspective-taking and effectiveness of discharge planning practices. Moreover, it serves as a guideline for developing educational programs for nurses to improve discharge planning practices.

## Supporting information

**S1 Checklist. STROBE statement.**
(DOCX)

**S1 Appendix. Draft questions generated by cognitive interview and I-CVI results (Round 1).** [a] The I-CVI (Item content validity index: I-CVI) was calculated by asking respondents to rate the degree to which each question item was related to the concept on a 4-point scale (1: not related to 4: fairly related), and the percentage of the number of respondents who answered 3 or 4 for each question was calculated. The first survey was 8 people rated the items, so items with an I-CVI of 0.75 or less were considered for modification.
(DOCX)

**S2 Appendix. Draft questions generated by cognitive interview and I-CVI results (Round 2).** Note: One nursing researcher dropped out from round 1. The order of the items was changed based on feedback from the initial cognitive interview. [a]The I-CVI (Item content validity index: I-CVI) was calculated by asking respondents to rate the degree to which each question item was related to the concept on a 4-point scale (1: not related to 4: fairly related), and the percentage of the number of respondents who answered 3 or 4 for each question was calculated. The second round was 7 people rated the items, so items with an I-CVI of 0.83 or less were considered for modification.
(DOCX)

**S1 Dataset. Supporting information data set (EFA, CFA, cor).**
(XLSX)

**S2 Dataset. Supporting information data set (ICC).**
(XLSX)

## Acknowledgments

I would like to express my gratitude to all those who cooperated in the research.

## Author Contributions

**Conceptualization:** Shingo Tanaka.

**Data curation:** Shingo Tanaka.

**Formal analysis:** Shingo Tanaka.

**Funding acquisition:** Shingo Tanaka.

**Investigation:** Shingo Tanaka.

**Methodology:** Shingo Tanaka, Masatoshi Saiki, Yukie Takemura.

**Project administration:** Shingo Tanaka.

**Supervision:** Yukie Takemura.

**Writing – original draft:** Shingo Tanaka.

**Writing – review & editing:** Masatoshi Saiki, Yukie Takemura.

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
