## [Decision Letter · Decision Letter 0]

26 Mar 2024

PONE-D-24-01714Development of the Ward Nurses’ Perspective-taking of the Staff Receiving Discharged Patients Scale: An Observational Study of Ward NursesPLOS ONE

Dear Dr. Tanaka,

Thank you for submitting your manuscript to PLOS ONE. After careful consideration, we feel that it has merit but does not fully meet PLOS ONE’s publication criteria as it currently stands. Therefore, we invite you to submit a revised version of the manuscript that addresses the points raised during the review process.

We look forward to receiving your revised manuscript.

Kind regards,

Fatma Ay, Ph.D

Academic Editor

PLOS ONE

Journal Requirements:

"This research was supported by the Graduate Program for Social ICT Global Creative Leaders (GCL) 

of The University of Tokyo by the Ministry of Education, Culture, Sports, Science and Technology

(MEXT), Japan."

"・ST.

・There is no grant number.

・The Global Creative Leaders (GCL) program of The University of Tokyo by the Ministry of Education, Culture, Sports, Science and Technology (MEXT).

・https://www.gcl.i.u-tokyo.ac.jp/faq-2/subsidies/

・The sponsors or funders did not play any role in the study."

4. In the online submission form, you indicated that "The participants of this study did not give written consent for their data to be shared publicly. Therefore, due to the sensitive nature of the research, supporting data is not available. The data that support the findings of this study are available from the corresponding author, Shingo Tanaka, upon reasonable request."

6. Please include a separate caption for each figure in your manuscript.

7. Please include your tables as part of your main manuscript and remove the individual files. Please note that supplementary tables (should remain/ be uploaded) as separate ""supporting information"" files

Reviewers' comments:

Reviewer's Responses to Questions

**Comments to the Author**

1. Is the manuscript technically sound, and do the data support the conclusions?

Reviewer #1: Partly

Reviewer #2: Yes

2. Has the statistical analysis been performed appropriately and rigorously? 

Reviewer #1: Yes

Reviewer #2: Yes

3. Have the authors made all data underlying the findings in their manuscript fully available?

Reviewer #1: Yes

Reviewer #2: Yes

4. Is the manuscript presented in an intelligible fashion and written in standard English?

Reviewer #1: Yes

Reviewer #2: Yes

5. Review Comments to the Author

Reviewer #1: Comments to the Author

Thanks to the authors for showing us the manuscript of “Development of the Ward Nurses’ Perspective-taking of the Staff Receiving Discharged Patients Scale: An Observational Study of Ward Nurses”. Overall, this manuscript is interesting. However, there are numerous concerns that need to be addressed.

1. This manuscript weakens the process of developing the scale, and only shows the verification results. The scale development process should be introduced in detail, and the scales obtained from each round of item selection, the items deleted and the final scale can be displayed in the appendix if there are too many contents.

2.The exploratory factor analysis method is commonly employed in scale development. Could you please clarify why the manuscript utilized both exploratory and confirmatory factor analysis methods simultaneously? Kindly provide an explanation within the manuscript.

3.The author should review the usage of technical terminology throughout the entire text, such as replacing "patient transfer" with "patient referral".

4.The full text format should be adjusted to align the two ends, enhancing its aesthetic appeal.

5.Keywords: The keywords should be separated by quotation marks, with each keyword capitalized at the beginning.

6.Abstract: The abstract is excessively verbose. It is advisable to be more succinct.

7.Introduction:

a)The introduction is excessively lengthy. I recommend condensing it logically and merging relevant theoretical content, with a focus on elaborating the significance of nurses' perspective in receiving discharged patients for institutional staff.

b)“It is defined as ‘the active cognitive process of imagining the world from another’s vantage point or imagining oneself in another’s shoes to understand their visual viewpoint, thoughts, motivations, intentions, and/or emotions’ [12 pp. 94-95]”, the utilization of “12pp” in this sentence lacks standardization.

c)“In this study, the staff receiving discharged patients comprise nurses, home care

nurses, and public health nurses in the jurisdictional district who receive patients from the hospital after discharge. ”，the paragraph commencing with this sentence introduces the purpose and significance within the preceding context, rendering it more suitable for placement in the methods section from both a content and structural standpoint.

8.Methods:

a)In the section of “Data collection”, the three subheadings “Item development”, “Evaluation by an expert”, and “Scale validation” did not appear to fall within the scope of data collection. Change is recommended.

b) The section on “Evaluation by an expert” lacks clear inclusion and exclusion criteria for experts, a specific process for evaluating the scale, indicators for making modifications, and a presentation of the obtained scale evaluation in this study.

c) The specific method of randomization used in the manuscript was not specified by the author, and no explanation of the process was provided. The authors are encouraged to provide a more detailed explanation of the randomization method employed and offer a comprehensive program description.

d) The section on "Surveyed facilities" lacks an explicit explanation in the text regarding the criteria and sources employed for selecting the 45 hospitals. It would be beneficial if the author could incorporate a comprehensive clarification within the text.

e) The “Participants” section should be organized and written in a manner that includes clear inclusion and exclusion criteria.

9.Results:

a)The low response rate of the questionnaire may result in sample underrepresentation, leading to bias and compromising the credibility of survey findings. With only 61 participants completing the second survey, it is necessary for the author to provide further explanations and clarifications in their manuscript.

b)The ICC value is unsatisfactory, and it is recommended to discuss in the discussion section for an analysis of the underlying reasons as well as exploration of potential improvement methods.

c)The results section should include a demonstration of the reliability and validity of the scale development process.

10.Discussion:

a) Authors are advised to incorporate a concise summary of the findings section within the introductory paragraph of the discussion section. And insert it into the first paragraph of this section.

b) As previously mentioned, this scale can assess the level of nurses' comprehension regarding the discharge plan staff, thereby enhancing inter-institutional collaboration and discharge plan effectiveness. The concept of perspective-taking originates from inherent human nature and falls within the realm of psychological research. While the impact of various motivations or intervention mechanisms may be more pronounced, this manuscript does not explicitly highlight the significance and value of this scale for discharge referral. Therefore, it is hoped that the author will provide an explanation in the discussion section.

c) "It differs from existing scales that measure the perspectives of other people and colleagues in daily life.", it lacks in-depth discussion on the comparison of scales, which measure the perspectives of other people and colleagues in daily life. I suggest providing specific explanations and conducting a thorough analysis to highlight the differences between this scale and existing ones, as well as exploring the underlying reasons for these distinctions.

Acknowledgments: “This research was supported by XXX”, the expression is not precision enough.

Tables:

a)In table 2, the specific explanation regarding F1 and F2 is currently unavailable.

b)The p-value annotation is not standardized and should be revised to p≤.

Reviewer #2: Thank you for the opportunity to review this excellent article. It is a well written manuscript both in terms of English and academic importance.

I hae a few minor comments which should be considered in order to improve the article.

1. Line 97 to 103 should be part of the methods section. The last sentence of the Background section should be: "the aim of this study...."

2. The limitations paragrahs require expansion, for example can a predominantly female sample also be valid for male nursing staff?

3. Is a response of 32-35% considered a high or acceptable response rate? In both cases of the questionnaire, time 1 and time 2, the response percentage was similar.

4. Line 164-165 Exclusion criteria mentioned - "nurse worked as nursing managers". Despite what is mentioned in the methods chapter, in table 1, it is written that about a third of the respondents to the first questionnaire are in management positions. Shouldn't they have been disqualified?

6. PLOS authors have the option to publish the peer review history of their article (what does this mean?). If published, this will include your full peer review and any attached files.

Reviewer #1: No

Reviewer #2: No

---

## [Author Response · Author response to Decision Letter 0]

21 Apr 2024

April 21, 2024

Fatma Ay

Academic Editor

PLOS ONE

Dear Dr. Ay:

Thank you for inviting us to submit our revised manuscript, “Development of the Ward Nurses’ Perspective-taking of the Staff Receiving Discharged Patients Scale: An Observational Study of Ward Nurses.”

We thank you and the reviewers for your thoughtful suggestions and insights. The manuscript has benefited from these insightful suggestions. I look forward to working with you and the reviewers to move this manuscript closer to publication in PLOS ONE.

We have incorporated a number of changes based on your suggestions, highlighted in yellow. We have also added new supplementary file based on the reviewer’s comments. To facilitate your review of our revisions, the following is a point-by-point response to the questions and comments in your letter dated April X, 2024.

Thank you for your consideration. I look forward to hearing from you.

Sincerely,

Shingo Tanaka

Department of Fundamentals of Nursing, School of Nursing, Yokohama City University, 3-9 Hukuura, Yokohama kanazawa-ku, Kanagawa 113-0033, Japan

Email: tanaka.4n5.res@gmail.com

Dear Editor

I made the following changes in order to adhere to the journal requirements: 

1. I have reviewed my paper to check if it fits the PLOS One style.

2. I have added a title page into main manuscript.

3. The Acknowledgements section in title page has been changed as follows: 

I would like to express my gratitude to all those who cooperated in the research.

Since no funding information is given in the main manuscript, I think there is no need to change the covering letter for now. 

4. I have revised data availability statement. 

5. I have verified that the ethics statements are not listed outside of the METHODS section.

6. I have included a separate caption and legend for each figure in my manuscript.

7. I have embedded tables into the main document.

8. I have added captions for supplementary files as an appendix.

[Reviewer(s)' Comments]

Reviewer #1

Thank you for your careful peer review. I have revised the manuscript based on your valuable suggestions. I will respond to each of your comments individually. The revised sections have been marked out in yellow highlight for your convenience. 

1. This manuscript weakens the process of developing the scale, and only shows the verification results. The scale development process should be introduced in detail, and the scales obtained from each round of item selection, the items deleted and the final scale can be displayed in the appendix if there are too many contents.

Thank you for your comment. In line with your suggestion, I have now included the proposed items from the item creation process in the appendix. Additionally, an explanation regarding the process of scale development has been added. For details, please review my response to the comments from you, particularly those related to the method.

2. The exploratory factor analysis method is commonly employed in scale development. Could you please clarify why the manuscript utilized both exploratory and confirmatory factor analysis methods simultaneously? Kindly provide an explanation within the manuscript.

Thank you for pointing this out. In order to clarify, I have added the following explanation to the " Determination of factor structure” section:

This scale was designed with a multidimensional structure in mind. Therefore, EFA was employed to identify and name the factors from the item pool. Subsequently, CFA was conducted to verify the construct validity by examining whether the model fit statistically across different samples.

3. The author should review the usage of technical terminology throughout the entire text, such as replacing "patient transfer" with "patient referral".

Thank you for pointing this out. I have replaced the term "patient transfer" with “discharge planning” in the key word section. I have reviewed previous studies and recognize that discharge planning is a more appropriate term for this study, as described in the main document definition.

4. The full text format should be adjusted to align the two ends, enhancing its aesthetic appeal.

Thank you for advice. I have made the necessary formatting changes.

5. Keywords: The keywords should be separated by quotation marks, with each keyword capitalized at the beginning.

The keywords have now been capitalized and are separated by quotation marks as suggested..

6. Abstract: The abstract is excessively verbose. It is advisable to be more succinct.

Thank you for your advice, I have summarized the Introduction and Conclusion in the Abstract as follows:

Introduction: Discharge planning involves coordinating between different care settings. Failed coordination can lead to delayed care at the facilities receiving discharged patients. Nurses who implement discharge planning must consider the circumstances of the staff receiving the discharged patients.

Conclusion: The newly developed scale proved to be reliable, valid, and suitable for use. This scale can measure the degree of perspective-taking by nurses, which can improve collaboration between facilities and the effectiveness of discharge planning.

7. Introduction:

a) The introduction is excessively lengthy. I recommend condensing it logically and merging relevant theoretical content, with a focus on elaborating the significance of nurses' perspective in receiving discharged patients for institutional staff.

Thank you for advice. In order to focus on perspective-taking, I have now condensed the paragraphs from the fifth paragraph onwards in the Introduction section.

b) “It is defined as ‘the active cognitive process of imagining the world from another’s vantage point or imagining oneself in another’s shoes to understand their visual viewpoint, thoughts, motivations, intentions, and/or emotions’ [12 pp. 94-95]”, the utilization of “12pp” in this sentence lacks standardization.

Thank you for pointing this out, I revised direct quotation style as following:

It is defined as “the active cognitive process of imagining the world from another’s vantage point or imagining oneself in another’s shoes to understand their visual viewpoint, thoughts, motivations, intentions, and/or emotions” [12] (94-95).

c) “In this study, the staff receiving discharged patients comprise nurses, home care nurses, and public health nurses in the jurisdictional district who receive patients from the hospital after discharge. ”，the paragraph commencing with this sentence introduces the purpose and significance within the preceding context, rendering it more suitable for placement in the methods section from both a content and structural standpoint.

Thanks for your suggestion. Based on the same, I have shifted the above text to the Method section as Definition.

8. Methods:

a) In the section of “Data collection”, the three subheadings “Item development”, “Evaluation by an expert”, and “Scale validation” did not appear to fall within the scope of data collection. Change is recommended.

Thank you for suggestion. I have now replaced the “Data collection” subheading with “Three steps of item development.” Additionally, I have added in the “Data collection of scale validation” section to provide information on the online survey.

b) The section on “Evaluation by an expert” lacks clear inclusion and exclusion criteria for experts, a specific process for evaluating the scale, indicators for making modifications, and a presentation of the obtained scale evaluation in this study.

Thank you for pointing out. I have included the items of each round of cognitive interview and I-CVI as an appendix, and added following explanation in main document:

Eight participants were evaluated in the first round and seven in the second round (one nurse researcher dropped out).

and

The inclusion criteria are participants with clinical experiences of at least 5 years as a nurse and in discharge planning. The exclusion criteria are only having clinical experience in the maternity ward.

In the cognitive interviews, based on their clinical experience, participants were asked about the parts of the questions that were difficult to understand and ones that should be added.

c) The specific method of randomization used in the manuscript was not specified by the author, and no explanation of the process was provided. The authors are encouraged to provide a more detailed explanation of the randomization method employed and offer a comprehensive program description.

Thank you for pointing out. I have now added the following explanation under “surveyed facilities”:

Only acute-care hospitals were extracted from the hospital databases provided by each local health authority [32–34], and 200 facilities were requested to participate in the survey, and were assigned random numbers.

d) The section on "Surveyed facilities" lacks an explicit explanation in the text regarding the criteria and sources employed for selecting the 45 hospitals. It would be beneficial if the author could incorporate a comprehensive clarification within the text.

As mentioned above, I have included the following explanation under “surveyed facilities”:

Only acute-care hospitals were extracted from the hospital databases provided by each local health authority [32–34], and 200 facilities were requested to participate in the survey, and were assigned random numbers.

e) The “Participants” section should be organized and written in a manner that includes clear inclusion and exclusion criteria.

Thank you for advice. This section has been revised as follows:

The inclusion criteria are ward nurses who had engaged in discharge planning. The exclusion criteria are nurses with less than a year of nursing experience, did not work in wards, worked in obstetric wards and unit care such as intensive care units and emergency wards, and worked as head ward nursing managers.

9. Results:

a) The low response rate of the questionnaire may result in sample underrepresentation, leading to bias and compromising the credibility of survey findings. With only 61 participants completing the second survey, it is necessary for the author to provide further explanations and clarifications in their manuscript.

Thank you for advice. From reviewing articles regarding discharge planning especially in Japan, I have determined that a response rate setting of about 20% is appropriate. As such, we have added the following explanations to the Method and Limitation sections, respectively:

(Method)

We performed sample size calculations to validate construct validity at time 1 and temporal stability at time 2. Regarding time 1, we found that 200 participants, 10 times the number of items in the draft scale, were needed based on the recommendation of Boateng et al. [22]. Regarding time 2, we found 19 participants were needed to calculate the intraclass correlation coefficients (ICC) between the first and the second surveys with 30 observed variables per participant (the null hypothesis was set at 0.5, the alternative hypothesis at 0.7, the power at 0.8, and the probability of significance at 0.05) [31]. The response rate was assumed to be about 20% based on previous discharge planning studies [32,33]. Therefore, it was determined that a distribution to approximately 1,000 persons was necessary for the analysis.

(Limitation)

The current study met the sample size requirements for time 1 and time 2 [22,31]. However, the response rate was low, similar to the response rate in a Japanese study of discharge planning. [32,33]. The low response rate may be due to the many questions and participants’ interest. In particular, a selection bias may have occurred if only nurses who expressed a strong interest in discharge planning were assumed to have responded to the study questions. 

b) The ICC value is unsatisfactory, and it is recommended to discuss in the discussion section for an analysis of the underlying reasons as well as exploration of potential improvement methods.

Thank you for the suggestion. I have added the following sentence to the Discussion section in the discussion around the reason for the ICC being low.

Further studies should be conducted on 1) how nurses’ perspective-taking of staff receiving patients changes over time, 2) whether an educational effect was seen in responding to the scale twice, and for how long this effect remains on the nurses.

Furthermore, errors in the ICC and paired t-test values have been corrected.

c) The results section should include a demonstration of the reliability and validity of the scale development process.

Thanks you for pointing this out." I have now created subsections for "scale validity" and "scale reliability" in order to revise the paper based on your suggestion..

10. Discussion:

a) Authors are advised to incorporate a concise summary of the findings section within the introductory paragraph of the discussion section. And insert it into the first paragraph of this section.

Thank you for advice. I added following sentences in first paragraph of Discussion section:

The EFA and CFA confirmed that the scale consists of two subfactors. Correlation analysis with other scales confirmed its construct validity, whereas Cronbach's alpha and ICC confirmed its reliability. However, the "imagine-self" subfactor was found to have low temporal stability.

b) As previously mentioned, this scale can assess the level of nurses' comprehension regarding the discharge plan staff, thereby enhancing inter-institutional collaboration and discharge plan effectiveness. The concept of perspective-taking originates from inherent human nature and falls within the realm of psychological research. While the impact of various motivations or intervention mechanisms may be more pronounced, this manuscript does not explicitly highlight the significance and value of this scale for discharge referral. Therefore, it is hoped that the author will provide an explanation in the discussion section.

Thank you for kind pointing out. I have now added following sentences in “scale availability” in Discussion section which serve as an explanation:

Prior research on nurses' perceptions focused on their interest in patients [20,21]. Additionally, studies on collaboration among medical staff focused on systems and the physical environment [54,55]. Therefore, the manner in which nurses' perceptions of the staff receiving discharged patients would affect their collaborations was unclear. The measures developed by this study focus on nurses’ recognition and contribute to clarifying the mechanism whereby the kind of event stimulates the perspective-taking and affects discharge planning.

c) "It differs from existing scales that measure the perspectives of other people and colleagues in daily life.", it lacks in-depth discussion on the comparison of scales, which measure the perspectives of other people and colleagues in daily life. I suggest providing specific explanations and conducting a thorough analysis to highlight the differences between this scale and existing ones, as well as exploring the underlying reasons for these distinctions.

Thank you for advice. I added following explanation to the “scale validity” sub-section in Discussion section:

It measures nurses’ perceptions of the resources available to the staff receiving discharged patients and the information they desire. Previous scales have measured perspective-taking toward friends in everyday life, staff in other departments of a company, and customers [11,17,18]. However, the scale developed in this study differs from other existing ones in that it measures perspective-taking toward staff in other facilities in the medical setting of discharge planning.

11. Acknowledgments: “This research was supported by XXX”, the expression is not precision enough.

Thank you for pointing this out. I have now revised the above as:

I would like to express my gratitude to all those who cooperated in the research.

12. Tables:

a) In table 2, the specific explanation regarding F1 and F2 is currently unavailable.

Thank you for checking. I have revised the terms from F1 and F2 to “Factor 1” and “Factor 2” in Table 2.

b) The p-value annotation is not standa

---

## [Decision Letter · Decision Letter 1]

23 May 2024

PONE-D-24-01714R1Development of the Ward Nurses’ Perspective-taking of the Staff Receiving Discharged Patients Scale: An Observational Study of Ward NursesPLOS ONE

Dear Dr. Tanaka,

Thank you for submitting your manuscript to PLOS ONE. After careful consideration, we feel that it has merit but does not fully meet PLOS ONE’s publication criteria as it currently stands. Therefore, we invite you to submit a revised version of the manuscript that addresses the points raised during the review process. Please submit your revised manuscript by Jul 07 2024 11:59PM. If you will need more time than this to complete your revisions, please reply to this message or contact the journal office at plosone@plos.org. Please include the following items when submitting your revised manuscript:A rebuttal letter that responds to each point raised by the academic editor and reviewer(s). You should upload this letter as a separate file labeled 'Response to Reviewers'.A marked-up copy of your manuscript that highlights changes made to the original version. You should upload this as a separate file labeled 'Revised Manuscript with Track Changes'.An unmarked version of your revised paper without tracked changes. You should upload this as a separate file labeled 'Manuscript'.

We look forward to receiving your revised manuscript.

Kind regards,

Fatma Ay, Ph.D

Academic Editor

PLOS ONE  

Reviewers' comments:

Reviewer's Responses to Questions

**Comments to the Author**

1. If the authors have adequately addressed your comments raised in a previous round of review and you feel that this manuscript is now acceptable for publication, you may indicate that here to bypass the “Comments to the Author” section, enter your conflict of interest statement in the “Confidential to Editor” section, and submit your "Accept" recommendation.

Reviewer #1: (No Response)

Reviewer #2: All comments have been addressed

2. Is the manuscript technically sound, and do the data support the conclusions?

Reviewer #1: Yes

Reviewer #2: Yes

3. Has the statistical analysis been performed appropriately and rigorously? 

Reviewer #1: Yes

Reviewer #2: I Don't Know

4. Have the authors made all data underlying the findings in their manuscript fully available?

Reviewer #1: Yes

Reviewer #2: Yes

5. Is the manuscript presented in an intelligible fashion and written in standard English?

Reviewer #1: Yes

Reviewer #2: Yes

6. Review Comments to the Author

**Reviewer #1: **Thanks to the authors for showing us the revised manuscript of “Development of the Ward Nurses’ Perspective-taking of the Staff Receiving Discharged Patients Scale: An Observational Study of Ward Nurses”. Overall, this manuscript is interesting. But there are plenty of concerns.

1.Please further refine the screening process. Firstly, provide a comprehensive explanation in the manuscript regarding the rationale behind setting the items adjustment criteria at 0.75 and 0.83 (I-CVI). Secondly, if we consider using 0.75 as the screening criterion in the initial round, it would necessitate removing 13 items. Please explain how the 28 items reviewed in the first round were refined to 22 items reviewed in the second round. Also, how the final 20 items were determined.

2.Keywords: The format of keywords still contains some errors, which should be revised to a more standardized format: Hospital nursing staff; Multidisciplinary care team.

3.Introduction:

a) The introduction section remains excessively lengthy and it is advisable to streamline it further.

b) “It is defined as ‘the active cognitive process of imagining the world from another’s vantage point or imagining oneself in another’s shoes to understand their visual viewpoint, thoughts, motivations, intentions, and/or emotions’ [12] (94-95)”, please explain the specific meaning of “(94-95) ”and elucidate it within the manuscript.

4. Methods: In the section of “Surveyed facilities”, while the authors included details of the randomization process for 200 hospitals, they did not clearly state the source of involvement for 45 hospitals. It is necessary to clarify whether these 45 hospitals have any affiliation with the aforementioned 200 hospitals.

**Reviewer #2: **(No Response)

7. PLOS authors have the option to publish the peer review history of their article (what does this mean?). If published, this will include your full peer review and any attached files.

Reviewer #1: No

Reviewer #2: No

---

## [Author Response · Author response to Decision Letter 1]

31 May 2024

June 1, 2024

Fatma Ay

Academic Editor

PLOS ONE

Dear Dr. Ay:

Thank you for inviting us to submit our revised manuscript, “Development of the Ward Nurses’ Perspective-taking of the Staff Receiving Discharged Patients Scale: An observational study of ward nurses.”

We thank you and the reviewers for your thoughtful suggestions and insights. The manuscript has benefited from these insightful suggestions. I look forward to working with you and the reviewers to move this manuscript closer to publication in PLOS ONE.

We have incorporated a number of changes based on your suggestions, highlighted in yellow. To facilitate your review of our revisions, the following is a point-by-point response to the questions and comments in your letter dated May 24, 2024.

Thank you for your consideration. I look forward to hearing from you.

Sincerely,

Shingo Tanaka

Department of Fundamentals of Nursing, Nursing Course, School of Nursing, Yokohama City University, 3-9 Hukuura, Yokohama kanazawa-ku, Kanagawa 113-0033, Japan

Email: tanaka.4n5.res@gmail.com

[Reviewer(s)' Comments]

Reviewer #1

Thank you for your careful peer review. I have revised the manuscript based on your valuable suggestions. I will respond to each of your comments individually. The revised sections have been marked out in yellow highlight for your convenience. 

Thanks to the authors for showing us the revised manuscript of “Development of the Ward Nurses’ Perspective-taking of the Staff Receiving Discharged Patients Scale: An Observational Study of Ward Nurses”. Overall, this manuscript is interesting. But there are plenty of concerns.

1. Please further refine the screening process. Firstly, provide a comprehensive explanation in the manuscript regarding the rationale behind setting the items adjustment criteria at 0.75 and 0.83 (I-CVI). Secondly, if we consider using 0.75 as the screening criterion in the initial round, it would necessitate removing 13 items. Please explain how the 28 items reviewed in the first round were refined to 22 items reviewed in the second round. Also, how the final 20items were determined.

Response: Thank you for pointing this out. The following explanations were added to address your comment:

The criteria for each round were calculated based on the recommendation of a previous study [29]. In both the rounds, items noted to have overlapping semantic content in the cognitive interviews were consolidated, and the use of subjects and predicates for each item was standardized. The researchers reviewed the items with I-CVI values below the criteria in the first round. Based on the results of the cognitive interviews, the items were rephrased, integrated with other items, or deleted, resulting in 22 items, down from the original 28. Items with I-CVI values that did not meet the criteria in the second round were similarly reviewed by the researchers and integrated into other items or deleted based on feedback from the cognitive interviews, resulting in 20 items from 22. The researchers reviewed the final draft to confirm the semantic content, and minor phrasing revisions were made to facilitate easy responses from participants.

2. Keywords: The format of keywords still contains some errors, which should be revised to a more standardized format: Hospital nursing staff; Multidisciplinary care team.

Response: Thank you for this suggestion. The format of the keywords has been revised to a standardized one. We checked with the guidelines and the editorial office and confirmed that there are no guidelines regarding keywords.

3. Introduction:

a) The introduction section remains excessively lengthy and it is advisable to streamline it further.

Response: Thank you for the kind advise. We have reorganized the paragraph introducing prior research on discharge planning.

However, we believe that the paragraphs introducing perspective-taking, considerations on the scale, and the significance of the study are essential for explaining the subsequent development.

b) “It is defined as ‘the active cognitive process of imagining the world from another’s vantage point or imaginingoneself in another’s shoes to understand their visual viewpoint, thoughts, motivations, intentions, and/or emotions’[12] (94-95)”, please explain the specific meaning of “(94-95) ”and elucidate it within the manuscript.

Response: Thank you for this comment. (94-95) refers to the page numbers of the referenced publication. We have ensured that the in-text citation is formatted according to the Vancouver style.

4. Methods: In the section of “Surveyed facilities”, while the authors included details of the randomization process for200 hospitals, they did not clearly state the source of involvement for 45 hospitals. It is necessary to clarify whether these 45 hospitals have any affiliation with the aforementioned 200 hospitals.

Response: Thank you for pointing this out. We have revised the text to indicate that 200 facilities were randomly selected from the database, and the 45 facilities that cooperated were included in the survey. 

Reviewer #2:

Thank you for your peer review. We have revised the manuscript in response to the other reviewer's comments. Please check our revised manuscript. We trust that the revisions have brought our manuscript closer to publication in PLOS ONE.

---

## [Decision Letter · Decision Letter 2]

11 Jun 2024

PONE-D-24-01714R2Development of the Ward Nurses’ Perspective-taking of the Staff Receiving Discharged Patients Scale: An Observational Study of Ward NursesPLOS ONE

Dear Dr. Tanaka,

Thank you for submitting your manuscript to PLOS ONE. After careful consideration, we feel that it has merit but does not fully meet PLOS ONE’s publication criteria as it currently stands. Therefore, we invite you to submit a revised version of the manuscript that addresses the points raised during the review process.

We look forward to receiving your revised manuscript.

Kind regards,

Fatma Ay, Ph.D

Academic Editor

PLOS ONE

Journal Requirements:

Reviewers' comments:

Reviewer's Responses to Questions

**Comments to the Author**

1. If the authors have adequately addressed your comments raised in a previous round of review and you feel that this manuscript is now acceptable for publication, you may indicate that here to bypass the “Comments to the Author” section, enter your conflict of interest statement in the “Confidential to Editor” section, and submit your "Accept" recommendation.

Reviewer #1: (No Response)

Reviewer #3: All comments have been addressed

2. Is the manuscript technically sound, and do the data support the conclusions?

Reviewer #1: Yes

Reviewer #3: Yes

3. Has the statistical analysis been performed appropriately and rigorously? 

Reviewer #1: Yes

Reviewer #3: Yes

4. Have the authors made all data underlying the findings in their manuscript fully available?

Reviewer #1: Yes

Reviewer #3: Yes

5. Is the manuscript presented in an intelligible fashion and written in standard English?

Reviewer #1: Yes

Reviewer #3: Yes

6. Review Comments to the Author

Reviewer #1: Comments to the Authors

Thanks to the authors for showing us the revised manuscript of “Development of the Ward Nurses’ Perspective-taking of the Staff Receiving Discharged Patients Scale: An Observational Study of Ward Nurses”. Overall, this manuscript is much better now. But there are still some questions.

1.During the construction of the professional scale, these items should be deleted if items falls below the pre-set standard (I-CVI). If these items are to be reintegrated and rephrased with other items, it is necessary to conduct a survey of the scale again before experts evaluation can be carried out.

2.Introduction:

a) In the section of introduction, 3 to 5 paragraphs primarily show the current research situation about “perspective-taking”, which is excessively lengthy. It is recommended to streamline and summarize the theoretical aspects instead of presenting literature details.

b) The Vancouver style employs superscript numbering and rarely includes the page number of the referenced literature. "12 pp. 94-95" requires further correction, and the entire text should be standardized to a similar format: [1].

Reviewer #3: It is a good topic for Nurse- mangers to apply this developed scale " Staff Receiving Discharged Patients Scale for Ward Nurses at hospitals.

The findings and results benefit all in-charge nurses to have a better plan to discharge the patients .

It is recommended to revise the whole paper by English language in Nursing sciences, It will be better.

7. PLOS authors have the option to publish the peer review history of their article (what does this mean?). If published, this will include your full peer review and any attached files.

Reviewer #1: No

Reviewer #3: No

---

## [Author Response · Author response to Decision Letter 2]

18 Jun 2024

June 18, 2024

Fatma Ay

Academic Editor

PLOS ONE

Dear Dr. Ay:

Thank you for inviting us to submit our revised manuscript, “Development of the Ward Nurses’ Perspective-taking of the Staff Receiving Discharged Patients Scale: An observational study of ward nurses.”

We thank you and the reviewers for your thoughtful suggestions and insights. The manuscript has benefited from these insightful suggestions. I look forward to working with you and the reviewers to move this manuscript closer to publication in PLOS ONE.

We have incorporated a number of changes based on your suggestions, highlighted in yellow. To facilitate your review of our revisions, we have prepared and attached herewith a point-by-point response to the questions and comments in your letter dated June 12, 2024.

Thank you for your consideration. We look forward to hearing from you.

Sincerely,

Shingo Tanaka

Department of Fundamentals of Nursing, Nursing Course, School of Nursing, Yokohama City University, 3-9 Hukuura, Yokohama kanazawa-ku, Kanagawa 113-0033, Japan

Email: tanaka.4n5.res@gmail.com

[Editor Comments]

Response: Thank you for raising this issue. We have updated the reference list in the revised manuscript. Additionally, we have checked whether each reference has been retracted by The Retraction Watch Database.

[Reviewer(s)' Comments]

Reviewer #1

Thank you for your careful peer review. We have revised the manuscript based on your valuable suggestions—the revised sections have been highlighted in yellow for your convenience. Furthermore, we have responded to each of your comments individually. 

Thanks to the authors for showing us the revised manuscript of “Development of the Ward Nurses’ Perspective-taking of the Staff Receiving Discharged Patients Scale: An Observational Study of Ward Nurses”. Overall, this manuscript is much better now. But there are still some questions.

1. During the construction of the professional scale, these items should be deleted if items falls below the pre-set standard (I-CVI). If these items are to be reintegrated and rephrased with other items, it is necessary to conduct a survey of the scale again before experts evaluation can be carried out.

Response: Thank you for pointing this out. Based on previous research, in the first round, we did not remove items without due consideration. However, we included a process to create revised items from feedback. We have revised the section “Evaluation by experts” to convey this.

2. Introduction:

a) In the section of introduction, 3 to 5 paragraphs primarily show the current research situation about “perspective-taking”, which is excessively lengthy. It is recommended to streamline and summarize the theoretical aspects instead of presenting literature details.

Response: Thank you for pointing this out. We have streamlined the Introduction based on your comment.

b) b) The Vancouver style employs superscript numbering and rarely includes the page number of the referenced literature. "12 pp. 94-95" requires further correction, and the entire text should be standardized to a similar format: [1].

Response: We agree with the reviewer’s advice and have revised the in-text citations and references as per the Vancouver style using numbers in square brackets according to the journal guidelines.

Reviewer #3:

It is a good topic for Nurse- mangers to apply this developed scale " Staff Receiving Discharged Patients Scale for Ward Nurses at hospitals.

The findings and results benefit all in-charge nurses to have a better plan to discharge the patients .

It is recommended to revise the whole paper by English language in Nursing sciences, It will be better.

Response: Thank you for pointing this out. We have had the revised manuscript checked by a professional proofreading service to ensure good language quality.

---

## [Editor Report · Decision Letter 3]

21 Aug 2024

Development of the Ward Nurses’ Perspective-taking of the Staff Receiving Discharged Patients Scale: An Observational Study of Ward Nurses

PONE-D-24-01714R3

Dear Dr. Tanaka,

We’re pleased to inform you that your manuscript has been judged scientifically suitable for publication and will be formally accepted for publication once it meets all outstanding technical requirements.

Kind regards,

Fatma Ay, Ph.D

Academic Editor

PLOS ONE
---

## [Editor Report · Acceptance letter]

9 Sep 2024

PONE-D-24-01714R3 

PLOS ONE

Dear Dr. Tanaka, 

I'm pleased to inform you that your manuscript has been deemed suitable for publication in PLOS ONE. Congratulations! Your manuscript is now being handed over to our production team.

Kind regards, 

on behalf of

Dr. Fatma Ay 

Academic Editor

PLOS ONE